# Cookie Consent Has Disparate Impact on Estimation Accuracy

**Erik Miehling**    **Rahul Nair**    **Elizabeth Daly**

**Karthikeyan Natesan Ramamurthy**    **Robert Redmond**

IBM Research

erik.miehling@ibm.com
{rahul.nair,elizabeth.daly}@ie.ibm.com
{knatesa,rredmond}@us.ibm.com

## Abstract

Cookies are designed to enable more accurate identification and tracking of user behavior, in turn allowing for more personalized ads and better performing ad campaigns. Given the additional information that is recorded, questions related to privacy and fairness naturally arise. How does a user's consent decision influence how much the system can learn about their demographic and tastes? Is the impact of a user's consent decision on the recommender system's ability to learn about their latent attributes uniform across demographics? We investigate these questions in the context of an engagement-driven recommender system using simulation. We empirically demonstrate that when consent rates exhibit demographic-dependence, user consent has a disparate impact on the recommender agent's ability to estimate users' latent attributes. In particular, we find that when consent rates are demographic-dependent, a user disagreeing to share their cookie may counter-intuitively cause the recommender agent to know more about the user than if the user agreed to share their cookie. Furthermore, the gap in base consent rates across demographics serves as an amplifier: users from the lower consent rate demographic who provide consent generally experience higher estimation errors than the same users from the higher consent rate demographic, and conversely for users who choose to withhold consent, with these differences increasing in consent rate gap. We discuss the need for new notions of fairness that encourage consistency between a user's privacy decisions and the system's ability to estimate their latent attributes.

## 1   Introduction

Increased regulation surrounding cookies has emerged in recent years in response to growing concerns over online privacy and data protection. In the EU, cookie policy is driven primarily by the General Data Protection Regulation (13) and the ePrivacy Directive (10) which dictate that if a website wishes to use cookies to identify and track users, beyond "strictly necessary cookies," then consent must be explicitly obtained from the user. There is currently no federal law regulating the use of cookies in the US, however, several states have imposed regulation that requires websites to disclose their data collection practices to users (5, 40).

Marketers use cookies to narrow in on target audiences most likely to buy their products, primarily via improved user tracking and enhanced personalization. The actual mechanisms for obtaining user consent can introduce some additional fairness and privacy concerns. Many cookie consent interfaces are designed such that web content is at least partially obscured until the user makes a

37th Conference on Neural Information Processing Systems (NeurIPS 2023).

consent decision. These interfaces often exhibit *deceptive design patterns* (38, 34, 15, 16) – design practices that incentivize users to make choices that lower their privacy. While some websites do offer a clear "reject all" button, many present the user with a decision between a simple "I agree" or "accept all" and a much less immediate choice of "manage preferences." This leads to many users agreeing to cookie tracking simply out of convenience (17). Furthermore, and an observation that forms the basis of the current paper, evidence suggests that users exhibit inherent differences in their likelihood of agreeing to cookie tracking. A recent study by YouGov (45) revealed that a user's consent rate is influenced by their age and culture/geographical location. Of the markets surveyed in the study, users ranged from 64% agree (Poland) to 32% agree (US), with older individuals being less likely to provide consent than younger individuals, perhaps partially due to their overall lowered trust in tech companies when it comes to their personal data (46).

In this paper, we investigate how demographic-dependent consent rates impact the ability of the recommender system to learn users' latent attributes and offer targeted recommendations. The main finding of our paper is that when user consent rates differ by demographics, the recommender system possesses disparate estimation accuracies across users in different demographics. The finding is based on a simple model of recommendations where, prior to the interaction, users provide cookie consent according to demographic-dependent probabilities. Users' consent decisions, along with the cookie for those who have provided consent, allow the recommender agent to form refined beliefs on users' demographics and preferences. Through sequential interaction with the users, the agent learns to personalize content to maximize engagement. We empirically illustrate the following:

1) *Disparate impact of consent:* Under demographic-dependent consent rates, users' consent decisions can have a disparate impact on the recommender agent's estimation accuracy of users' latent attributes. In particular, we find that withholding consent can lead to *lower* estimation errors for users in the lower consent rate population. In other words, a recommender system may know more about a person who opts not be tracked compared to those who willingly provide tracking information. Additionally, users from a population with a low consent rate who provide consent experience higher estimation errors than the same users from a high consent rate population.

2) *Amplification effects:* The difference in consent rates across cohorts serves as an amplifier for the above effects, with the disparities in estimation errors across cohorts increasing in the consent rate gap.

Note that our model is simple: ads are described by a single feature, the user pool is fixed, and user affinities/preferences do not change over time. This simplicity is intentional, with the goal being to isolate and understand the effects of specific model aspects (i.e., impact of different user consent rates) without the additional noise and confounding factors present in real-world settings. The observations on disparate impact that emerge from our setting should serve as an indicator that the same negative results may also exist in more complex recommender systems. Additionally, given the fundamental nature of the observed disparities, we believe that the search for mitigation strategies in this simplified setting can provide useful insights for the design of more complex systems.

## 2 Background

**Recommender systems.** The main challenge in recommender systems is making recommendations when only a sparse set of preference data is available. Broadly, recommender systems address this missingness via two approaches: content-based filtering, and collaborative filtering.[1] Content-based filtering methods use known information about users and items to suggest content. For example, a content-based movie recommender system may compare known features of a movie (degree of humor, action, drama) with known user preferences to suggest movies to users that yield the greatest similarity. Collaborative filtering, on the other hand aims to *learn* these relevant features based on patterns in the observed preference/response data. Items in a collaborative filtering-based system are recommended to users based on what content other users with similar behavior have consumed (hence the term collaborative).

---

[1]Hybrid approaches, as well as methods based that leverage more sophisticated learning methods, are also popular. For a comprehensive review see (4).

A popular collaborative-filtering algorithm is matrix factorization. Matrix factorization asserts that user data is not missing uniformly at random, but rather according to some low rank structure of which it aims to discover. The learned factors can then be used to infer unobserved ratings for other user-item pairs, in turn driving recommendations. Associating each user with factor $u_i \in \mathbb{R}^k$ and each ad with factor $v_a \in \mathbb{R}^k$, latent factor estimates $(\hat{\boldsymbol{u}}, \hat{\boldsymbol{v}}) = ((u_i)_i, (v_a)_a)$ are obtained under standard matrix factorization via solution of a regularized loss-minimization on the known user responses

$$(\hat{\boldsymbol{u}}, \hat{\boldsymbol{v}}) = \underset{(\boldsymbol{u}, \boldsymbol{v})}{\operatorname{argmin}} \left[ \sum_{(i,a) \in \mathcal{R}} (u_i^\top v_a - r_{i,a})^2 + \lambda \left( \sum_i \|u_i\|_2 + \sum_a \|v_a\|_2 \right) \right]$$

where $\mathcal{R}$ is the set of user-item pairs $(i, a)$ where a response $r_{i,a}$ has been recorded. A variety of modifications of standard matrix factorization exist (25), including incorporating ranking biases, user and item side information, varying confidence weights, and temporal effects. The above optimization is typically carried out on a mix of both historical data and new interaction data, with the latter data incorporated periodically via retraining.

**Cookies, consent, and behavioral advertising.** Cookies are small pieces of data that are stored on a user's device with the specific purpose ranging from storing login credentials (first-party cookies) to tracking user behavior for targeted advertising (third-party cookies). Given their capability of enabling inference of potentially sensitive user information (9, 39), regulation has emerged to increase transparency around their use. While specific laws vary, many countries now require web publishers to present some form of banner or pop-up either informing the user that information is being recorded or requesting the user to indicate their cookie consent decision prior to browsing the page.

Behavioral advertising describes the practice of using learned user behavior (e.g., user preferences) to make personalized recommendations. Cookies help to facilitate inference of this information via tracking user behavior across websites (e.g., search history, clicks, purchases). The end goal of behavioral advertising is to increase relevancy of recommendations, in turn increasing user engagement and the return of the ad campaign.

## 3 Related work

**Algorithmic feedback loops.** Much of the analysis of algorithmic feedback loops in the literature is centered on recommender systems and the trade-off between recommendation accuracy and topic diversity. It is generally accepted that as the recommender system learns to generate high engagement recommendations, topic diversity suffers which in turn may cause filter bubbles and echo chambers to emerge (51, 26, 33, 22, 6, 27, 24, 11, 18, 30, 50, 14). While a variety of mitigation techniques have been proposed – including techniques to identify and remove these effects (37), slow degeneracy (24), improve user heterogeneity (43), and disentangle user interest from user conformity (49) – the question of how to best balance accuracy with diversity, and addressing broader fairness concerns, is still very much an active area of research.

The above papers illustrate the many potential downsides of engagement-driven recommendation and collaborative filtering. To the best of the authors' knowledge, our paper is the first to analyze the impact of consent rates on the dynamics of the recommendation process.

**Privacy, fairness, and bias considerations in recommender systems.** Given that recommender systems are designed to personalize content to users, questions related to privacy, fairness, and bias naturally arise (23, 12, 7). Many notions of recommender system bias have been studied in the literature including *participation inequality* where system usage varies across a user's attributes (e.g., gender, race, language, etc.), *selection bias* where the users' choice behavior leads to non-representative item feedback, *conformity bias* where users in similar groups act similarly, *exposure bias* where suppliers/items exhibit fundamentally different levels of visibility, and *popularity bias* where popular items are recommended more often than long tail items, among various others. Furthermore, it is widely recognized that the inherent feedback nature of recommender systems amplifies these biases (6, 30).

Research on the privacy and fairness of recommender systems aims to gain a deeper understanding of how the structure of the recommender system influences these biases and subsequently propose mitigation strategies via modified recommendation algorithms, with a large thread of research concerning the design of privacy-preserving recommender systems (1, 31, 47, 3). Most related to our

paper are the structural questions of characterizing the ability of the recommender system to recover protected attributes from ratings of both a homogeneous user pool (42) and a mix of public/private users (44). Notably, (44) demonstrates that only a small number of public users (users willing to share preference information) with a large number of ratings in a pool of private users (users with hidden preferences) is sufficient to produce accurate system-wide recommendations. In contrast, our paper studies how inherent differences in user consent rates influences the system's ability to estimate users' latent attributes across groups.

**Simulators for recommender systems.** Given the complexity of the interactions between users and the learning behavior of the agent, simulators have become an increasingly popular tool for understanding the dynamics of recommender systems. Agent-based modeling allows for simulation of fine-grained user interactions, facilitating useful insights without running costly field tests. A variety of open-source recommender system simulators have emerged in recent years, namely `RecoGym` (35), `RecSim` (21), `PyRecGym` (36), `Surprise` (20), `RecSim NG` (32), and others (48), (8). The simulator developed for purposes of this study augments `RecSim` with the ability for the learning agent to maintain asymmetric (Bayesian) uncertainty over users.

## 4 A Simple Recommendation Model with Cookie Consent

Consider a recommendation environment consisting of a recommender, termed the *agent*, sequentially interacting with a fixed population of $n$ individuals or *users* (see Fig. 1). Before the interaction begins, users make consent decisions according to known cohort-dependent probabilities. The agent uses the consent decisions and revealed cookies to form refined (interim) beliefs on the users' cohorts, which in turn guide recommendations. The agent is periodically retrained using the updated history of recommendation-response pairs, with the overall goal of recommending content that maximizes engagement.

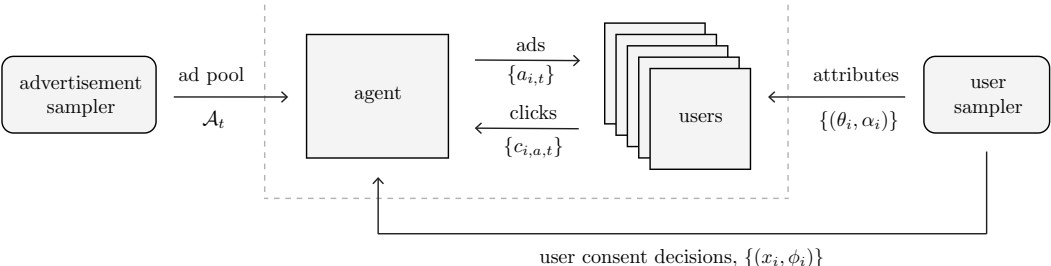

Figure 1: The recommendation process. The advertisement sampler generates the ad pool (assumed to be refreshed in each round $t$). The user sampler initializes both latent attributes of users and consent decisions. In each round, the agent generates recommendations (ads) and records user responses (clicks).

**Advertisement model.** Each ad is described by a single *topic* feature $\tau \in \mathcal{T} = \{\tau^1, \ldots, \tau^m\}$, where $m$ is the number of possible ad topics. In each round $t$, a subset of $l < m$ ad topics are sampled to form the current set of recommendable ads $\mathcal{A}_t$, termed the *ad pool*.

**User model.** Each user $i \in [n]$ is described by three features: a *cookie* $\phi_i \in \Phi = \{\phi^1, \ldots, \phi^c\}$, a demographic or *cohort* $\theta_i \in \Theta = \{\theta^1, \ldots, \theta^d\}$, and a vector of *topic affinities* $\alpha_i \in \mathbb{R}_+^m$. Each user's cookie-cohort pair is drawn from a known joint prior $\mu \in \Delta(\Phi \times \Theta)$, i.e., knowledge of a user's cookie is at least partially informative for its cohort. Consent decisions are dictated by cohort-dependent probabilities $q_\theta$, $\theta \in \Theta$, with each user $i$ in cohort $\theta$ revealing its cookie with the agent according to $X_i \sim \text{Bernoulli}(q_\theta)$.[2] Topic affinities are generated according to log-normal distributions, with each user $i$ from a given cohort $\theta$ possessing affinities $\alpha_i \sim \text{LogNormal}(\rho_\theta, \sigma_\theta^2)$, where $\rho_\theta \in \mathbb{R}^m$ and $\sigma_\theta \in \mathbb{R}_+^m$ are the cohort-dependent means and standard deviations of $\alpha_i$'s natural logarithm.

---

[2]Note that we model every user as possessing an underlying cookie, with the agent only being aware of the user's cookie if the user decides to share, i.e., $x_i = 1$.

In each round $t$, each user is faced with a single ad and makes a binary decision to either click or not click. Specifically, when user $i$ is recommended an ad $a$, the user first scores the ad via the utility function $u(a; \alpha_i) = \alpha_i^\top \mathbb{1}_{\tau_a}$ where $\mathbb{1}_{\tau_a} \in \mathbb{R}^m$ is the indicator vector on topics (a vector of zeros with a one in location of the topic of ad $a$). User $i$'s choice to click on $a$, denoted by $c_{i,a} \in \{0,1\}$, is stochastic and is dictated by $c_{i,a} \sim \text{Bernoulli}(p_{i,a})$, where $p_{i,a}$ is the click probability given by the following logit model

$$p_{i,a} = \frac{\exp(u(a; \alpha_i))}{\exp(u_0) + \exp(u(a; \alpha_i))} \tag{1}$$

where $u_0$ is the *no-click-mass* used to model the possibility of the user not clicking on $a$, assumed to be homogenous across users.

**Recommender agent.** The agent faces a fundamental trade-off between recommending high-engagement content and learning more about users' tastes. Our recommendation model combines two foundational models from the literature, namely incorporation of confidence weights (19, 25) into an online matrix factorization procedure (41). Online matrix factorization addresses the conflicting objectives of the agent – exploration to improve estimates of latent factors, and exploitation of known high engagement content – by interleaving estimation of the latent factors with specification of recommendations. Including confidence weights allow us to model the agent's heterogeneous uncertainty on users due to their consent decisions and personalized recommendations. Note that while this is a stylized recommender model, it possesses the main features seen in more complex systems, specifically online learning and heterogeneous uncertainty across users.

Formally, the recommendation process evolves as follows. Given the agent's prior belief, $\mu \in \Delta(\Phi \times \Theta)$, the users' consent decisions are used to form interim cohort beliefs $\tilde{\mu}_i \in \Delta(\Theta)$ for each user $i$. If user $i$ gives consent ($x_i = 1$) then the agent is informed of the user's cookie, $\phi_i$, and its updated belief of the user's cohort is formed as $\tilde{\mu}_i(\vartheta) = p(\vartheta \mid x_i = 1, \phi_i) = \frac{q_\vartheta \mu(\phi_i, \vartheta)}{\sum_{\vartheta'} q_{\vartheta'} \mu(\phi_i, \vartheta')}$ for each $\vartheta \in \Theta$. If user $i$ does not provide consent ($x_i = 0$), no cookie information is revealed and the agent's belief of the user's cohort is given by $\tilde{\mu}_i(\vartheta) = p(\vartheta \mid x_i = 0) = \frac{(1-q_\vartheta) \sum_\varphi \mu(\varphi, \vartheta)}{\sum_{\vartheta'} (1-q_{\vartheta'}) \sum_{\varphi'} \mu(\varphi', \vartheta')}$, $\vartheta \in \Theta$. Interim beliefs $\tilde{\mu} = (\tilde{\mu}_1, \ldots, \tilde{\mu}_n)$ are used to form beliefs $\mu_0 = (\mu_{0,1}, \ldots, \mu_{0,n})$ via an offline response set $\mathcal{L}_0$ of recommendation-response pairs of the form $\{(a_i, c_{i,a})\}$ across users.

The recommender model at round $t$ is represented by a pair of user-ad latent factor estimates $(\hat{\boldsymbol{u}}_t, \hat{\boldsymbol{v}}_t)$. Recommendations are generated via an $\varepsilon$-greedy bandit: with probability $\varepsilon$, the ad recommended to user $i$ in round $t$, denoted by $a_{i,t}$, is chosen uniformly at random, $a_{i,t} \sim \text{U}(\mathcal{A}_t)$, and with probability $1 - \varepsilon$, the recommendation is the ad in the current ad pool with the highest estimated value $a_{i,t} = \operatorname{argmax}_{a \in \mathcal{A}_t} \hat{u}_{i,t}^\top \hat{v}_{a,t}$. Responses to the recommendations are appended to the response set, $\mathcal{L}_{t+1} = \mathcal{L}_t \cup \{(a_i, c_{i,a}) \mid i \in [n], a_i \in [m]\}$, and are used to maintain a cumulative count of clicks $r_{i,a,t}$ across user-ad pairs.

Retraining consists of updating cohort beliefs and latent factor estimates using recommendation-response pairs, and is performed every $T_b$ rounds. If $t$ is a retraining round, cohort beliefs are first updated according to a Bayesian update $\mu_{i,t} = f(\mu_{i,t-1}, \mathcal{L}_t \setminus \mathcal{L}_{t-T_b})$ where $\mathcal{L}_t \setminus \mathcal{L}_{t-T_b}$ is the set of responses since the previous retraining round. The heterogeneous beliefs on user cohorts gives rise to a weighted procedure. For a given set of cohort beliefs $\mu_t$, the agent computes confidence weights $\bar{w}_t = (\bar{w}_{i,a,t})_{i,a}$ as the expected probability for seeing the current response counts. Specifically, each weight $\bar{w}_{i,a,t}$ is the expected binomial probability, given $\mu_t$, for seeing $r_{i,a,t}$ defined as

$$\bar{w}_{i,a,t} = \mathbb{E}_{\vartheta_i \sim \mu_{i,t}}[p_t(I_{i,a,t}, r_{i,a,t}, \vartheta_i)]$$

where $p_t(I, r, \vartheta)$ is the binomial probability at round $t$ given impressions $I$, positive response counts $r$, and cohort $\vartheta$. Updated latent factor estimates are computed via a confidence-weighted matrix factorization procedure as

$$(\hat{\boldsymbol{u}}_t, \hat{\boldsymbol{v}}_t) = \operatorname*{argmin}_{(u,v) \in \mathcal{U} \times \mathcal{V}} \left[ \sum_{(a_i, c_{i,a}) \in \mathcal{L}_t} \bar{w}_{i,a,t}(u_i^\top v_a - r_{i,a,t})^2 + \lambda \left( \sum_{i \in [n]} \|u_i\|_2 + \sum_{a \in [m]} \|v_a\|_2 \right) \right] \tag{2}$$

where $\mathcal{U} = \{(u_1, \ldots, u_n) \mid u_i \in \mathbb{R}^k, i \in [n]\}$, $\mathcal{V} = \{(v_1, \ldots, v_m) \mid v_a \in \mathbb{R}^k, a \in [m]\}$ are the latent factor spaces, $\mathcal{L}_t$ is the current history of responses, and $\lambda > 0$ is a regularization weight. If $t$ is not a retraining round, then cohort beliefs and latent factor estimates are propagated as is from the previous round. Expressions for the Bayesian updates and expected binomial probabilities can be found in Appendix A, with pseudocode of the recommendation process in Appendix B.

# 5 Experiments

The following experiments study the dynamics of the recommendation process described in Section 4 particularly as it relates to users' consent rates. Empirical results were obtained via a simulator based on RecSim (21). Full details of the simulator and the experimental setup can be found in the supplementary material (Appendix C). Base model parameters assumed throughout this section are: number of users $n = 1000$, number of ads $m = 200$, ad pool size $l = 50$, and number of cohorts $d = 2$. Sensitivity analyses can be found in Appendix D.

The following notation/terminology will be used throughout this section to aid explanations. Let $N_1^\vartheta = \{i \in [n] \mid x_i = 1, \theta_i = \vartheta\}$ denote the set of users in cohort $\vartheta$ who have decided to share their cookie, with $N_0^\vartheta$ defined analogously. We refer to $N_1^\vartheta$ as the *consent group* (in cohort $\vartheta$) and $N_0^\vartheta$ as the *non-consent group*. Estimation errors on cohorts for each group are quantified by log loss, i.e., for a set of users $N$ in group $(\vartheta, x)$, the (average) log loss is denoted by $L_x^\vartheta(\mu) = -\frac{1}{|N|} \sum_{i \in N} \sum_{\vartheta \in \Theta} \mathbf{1}(\theta_i = \vartheta) \log(\mu_i(\vartheta))$. Estimation errors on topic affinities are given by MSE.

## 5.1 Impact of consent on estimation accuracy

The following set of experiments investigates the impact of cohort-dependent consent rates on the agent's estimates of the users' cohorts and topic affinities. Given users' consent decisions, and cookies for those who decided to provide consent, the agent forms interim beliefs on each user, $\tilde{\mu}_i$, $i \in [n]$, as per the model description in Section 4. Fig. 2 illustrates the impact of these consent decisions on cohort and affinity estimation errors, for $\Theta = \{\theta, \theta'\}$, in two cases: homogeneous consent rates (Fig. 2(a)) and heterogeneous consent rates (Fig. 2(b)).

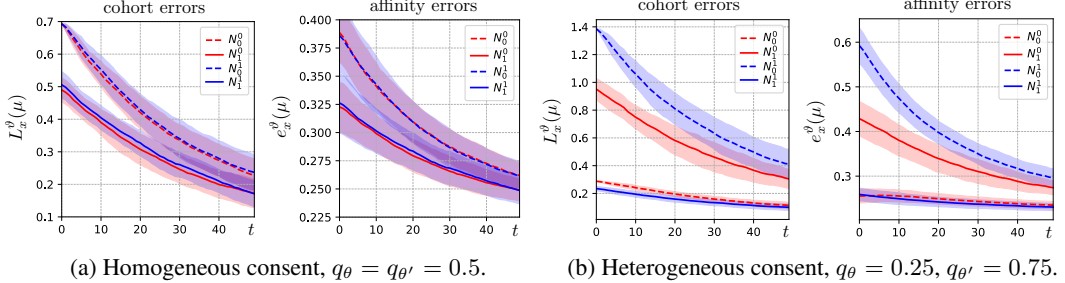

(a) Homogeneous consent, $q_\theta = q_{\theta'} = 0.5$.  (b) Heterogeneous consent, $q_\theta = 0.25$, $q_{\theta'} = 0.75$.

Figure 2: Estimation errors over training rounds $t$ $(T_b = 1)$ as a function of consent rates under binary cookie space $\Phi = \{\phi, \phi'\}$ and symmetric prior $\mu(\phi, \theta) = \mu(\phi', \theta') = 0.4$.

Fig. 2(a) illustrates that when users' consent rates are identical across cohorts, estimation errors for the non-consent group $N_0^\theta$ are higher than the errors for the consent group $N_1^\theta$ regardless of their cohort $\theta$.

As seen in Fig. 2(b), this intuitive result breaks down when users' consent rates differ across cohorts. While withholding consent leads to a higher estimation error for the higher consent rate group, $\theta'$, withholding consent can lead to a *lower* estimation error for the lower consent rate group, $\theta$. This inversion effect persists for the lower consent rate population for any sufficiently large separation of consent rates across cohorts, as illustrated by Fig. 3. These cohort-dependent effects of consent are summarized by Observation 1.

**Observation 1** (Comparison within cohorts)**.** *Let $\Theta = \{\theta, \theta'\}$ and let $\mu$ be any non-fully informative prior. If $q_\theta = q_{\theta'}$ (consent rates are independent of cohort) then non-consent leads to a higher estimation error than consent. If $q_\theta < q_{\theta'}$ (consent rates are cohort-dependent with users in cohort $\theta$ providing consent as a lower rate than users in cohort $\theta'$), then:*

    *a. There exists a (prior-dependent) constant $\delta_\mu$ such that for $q_{\theta'} - q_\theta > \delta_\mu$, the non-consent group in cohort $\theta$, $N_0^\theta$, experiences lower estimation errors than the consent group, $N_1^\theta$.*

    *b. The non-consent group in cohort $\theta'$, $N_0^{\theta'}$, experiences higher estimation errors than the consent group, $N_1^{\theta'}$.*

Observation 1 holds for any non-fully informative prior, meaning that there exists at least one cookie that does not reveal the user's cohort with certainty – a property that is almost certainly satisfied in practical (i.e., noisy) recommender systems.

For partially informative priors (each cookie only partially reveals the user's cohort), the statement of the above observation can be understood by looking at the agent's sources of information. Consider the following signals: i) the cookie value in the event that the user provided consent and, ii) under heterogeneous consent rates, the consent decision itself. The interplay between these two sources of information means that, under some priors, the act of a user withholding consent can itself be more informative than providing consent and revealing their cookie. Fig. 3 illustrates the estimation errors for different consent rate regimes under symmetric prior $\mu(\theta, \phi) = \mu(\theta', \phi') = 0.45$.

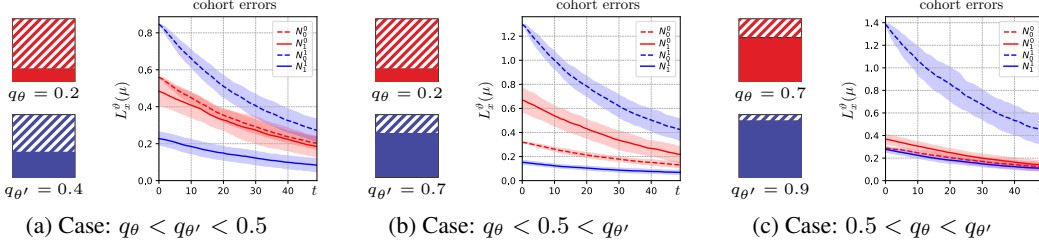

(a) Case: $q_\theta < q_{\theta'} < 0.5$      (b) Case: $q_\theta < 0.5 < q_{\theta'}$      (c) Case: $0.5 < q_\theta < q_{\theta'}$

Figure 3: Impact of consent versus non-consent on the agent's cohort estimation error in each cohort for various consent rate regimes under symmetric prior $\mu(\phi, \theta) = \mu(\phi', \theta') = 0.45$.

The above figure, specifically Fig. 3(a), illustrates that under a very informative prior, a small gap in consent rates, $(q_\theta, q_{\theta'}) = (0.2, 0.4)$, yields a consent signal that contains relatively low informativeness compared to the revealed cookie value, causing consent to yield lower estimation errors for both cohorts. However, as the absolute consent rates increase in both cohorts, this same consent rate gap (compare Figs. 3(a) and (c)) yields a consent signal that is sufficiently informative to yield lower estimation errors for withholding consent for the lower consent rate group.

Generally the larger the gap in consent rates, the more information the consent signal carries for the user's cohort. Apart from very informative priors, the consent signal carries sufficiently rich information to yield lower estimation errors for withholding consent for the lower consent rate population even for small gaps in the consent rates.

Observation 1 compares the impact of consent decisions within a given cohort under homogeneous and heterogeneous consent rates. A comparison of the impact of the same consent decision across users in different cohorts is summarized by the following observation.

**Observation 2** (Comparison across cohorts). *Let $\Theta = \{\theta, \theta'\}$ and let $\mu$ be any partially informative prior. If $q_\theta = q'_\theta$, then the consent group experiences the same estimation errors regardless of cohort (similarly for non-consent). If $q_\theta < q_{\theta'}$, then:*

a. *There exists a (prior-dependent) constant $\eta_\mu$ such that for $q_{\theta'} - q_\theta > \eta_\mu$, the non-consent group in cohort $\theta'$, $N_0^{\theta'}$, experiences higher estimation errors than the non-consent group in cohort $\theta$, $N_0^\theta$.*

b. *The consent group in cohort $\theta'$, $N_1^{\theta'}$, experiences lower estimation errors than the consent group in cohort $\theta$, $N_1^\theta$.*

## 5.2 Amplification effects

The following experiments investigate how the disparities observed in (O.1) and (O.2) are influenced by the relative consent rates across cohorts. For $\Theta = \{\theta, \theta'\}$, define $\delta_t^\vartheta = \bar{L}_0^\vartheta(\mu_t) - \bar{L}_1^\vartheta(\mu_t)$ as the difference in mean cohort errors between non-consent and consent for users in cohort $\vartheta \in \Theta$. Additionally, define $\gamma_t^x = \bar{L}_x^{\theta'}(\mu_t) - \bar{L}_x^\theta(\mu_t)$ as the difference in mean cohort errors for consent decision $x$.

In words, the quantities $\delta_t^\vartheta$, $\vartheta \in \Theta$, represent how much worse the agent's estimation error is under non-consent versus consent for users in cohort $\vartheta$. The quantity $\gamma_t^0$ (resp. $\gamma_t^1$) represents how much higher the agent's estimation error is for a user who withheld (resp. provided) consent if they belonged to cohort $\theta'$ versus if they belonged to cohort $\theta$. Figs. 4 and 5 illustrate how these quantities differ as a function of consent rates for a given number of training rounds ($\tau = 10$).

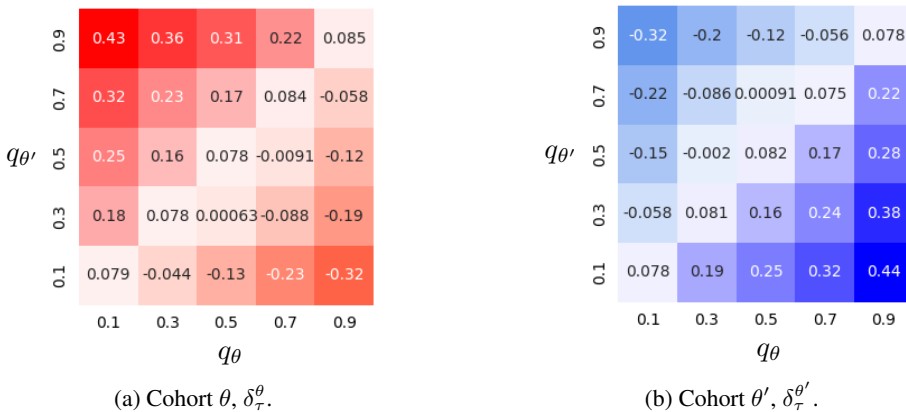

(a) Cohort $\theta$, $\delta_\tau^\theta$.               (b) Cohort $\theta'$, $\delta_\tau^{\theta'}$.

Figure 4: Relative errors for non-consent versus consent as a function of consent rates $(q_\theta, q_{\theta'})$ for each cohort and fixed number of training rounds $\tau = 10$.

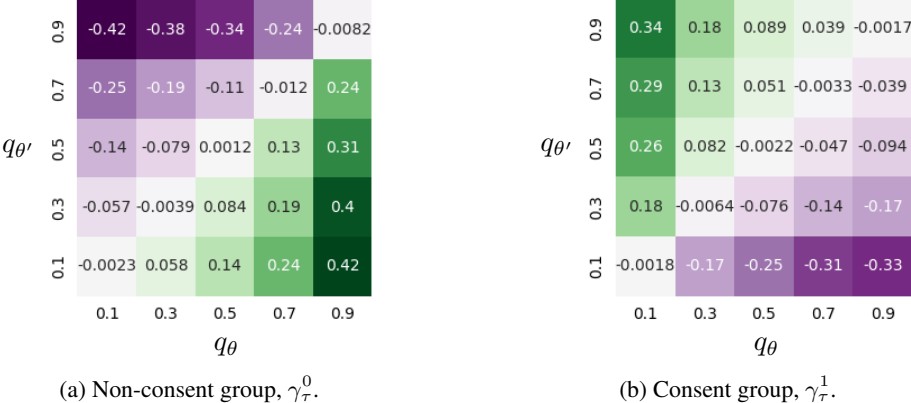

(a) Non-consent group, $\gamma_\tau^0$.               (b) Consent group, $\gamma_\tau^1$.

Figure 5: Relative errors for a given consent decision across cohorts as a function of consent rates $(q_\theta, q_{\theta'})$ for fixed number of training rounds $\tau = 10$.

Fig. 4 illustrates that the relative loss in accuracy for non-consent (compared to consent) is greatest in cohort $\theta$ when $q_\theta$ is minimal and $q_{\theta'}$ is maximal, with the opposite effects observed in cohort $\theta'$. Fig. 5 illustrates that the users who withhold consent are on average impacted greater by a given gap in consent rates than users who consent, e.g., compare the magnitude of the values at $(q_\theta, q_{\theta'}) = (0.25, 0.75)$ for non-consent versus consent.

In summary, the gap in consent rates serves as an amplifier for an individual user's consent decision. Users from the higher consent rate cohort who withhold consent experience higher estimation errors than the same users from the lower consent rate cohort, with the opposite holding for users in the consent group.

# 6   Discussion

Our results demonstrate that even in a simple recommendation model with reasonable modeling assumptions, the impact of a user's cookie consent decision is not necessarily straightforward. Particularly, when consent rates exhibit demographic-dependence, the user's consent decision itself acts as a signal, carrying information beyond the information contained in the actual cookie value. This leads to a demographic-dependent impact of consent, resulting in a user's privacy-protective decision (of disagreeing to share their cookie) potentially causing the system to know more about them than if the user were to intentionally reveal their cookie information.

To draw a comparison with the large body of existing literature in the fairness of recommender systems, many of the existing papers (see the papers referenced in Section 3) draw attention to the way in which the system learns similarities among users over the recommendation process as a primary source of unfairness. Indeed, this is fundamental to how matrix factorization algorithms (and many more complex recommender systems) operate: users who behave similarly, i.e., how they respond to the same recommendations, will be deemed as being similar (even falsely so). This will lead to these users seeing similar recommendations which can, in turn, lead to undesirable outcomes.

However, one of the goals of our paper is to draw attention to the informational effects of consent as a potential additional source of unfairness. When consent is a conscious decision of the user, and the rate of a given consent decision differs based on sensitive attributes of the user, e.g., age, then the system can use the consent decision to learn more about the user's sensitive attributes. When embedded within an algorithmic system, platforms may unintentionally profile users based on the absence of data.

These observations have possible implications in the design of recommender systems. Any information that the system can use to draw similarities will be used, most notably how users respond to recommendations, but also, and the focus of our paper, the inherent likelihood of a user to provide consent to cookie sharing. These similarities are used to infer to which group a user belongs and subsequently drive recommendations. The implications for algorithm design is that one must be mindful of the complexities of information revelation especially as it relates to how the algorithm will use the revealed information.

**New notions of fairness.** The design of recommender systems that are more aware of the informational effects of users' decisions may require new notions of fairness. It is well-known in the fairness community that static fairness metrics are insufficient for dynamic settings, and can actually actively harm the group they intend to protect (28).

In our setting, whether the system possessing heterogeneous accuracy about users results in a fair or unfair outcome depends on the product being recommended. Given our observations that the specific impact of a user's consent decision varies depending on the user's demographic, it is not hard to see that such accuracy disparity may lead to unfair situations. For example, consider an ad recommendation setting where some of the ads may be predatory to older individuals. According to our model, an older individual's desire to remain more private by deciding to disagree to cookie sharing may allow the system to know more about them, simply by virtue that more older individuals choose to not share their cookies, and subsequently make them a more likely target for such ads. Consequently, we emphasize the need for development of fairness metrics that ensure consistency of an individual user's privacy-protective decision and the amount (or accuracy) of information that the system knows about that user.

It is worth noting that this informational consistency property is distinct from the literature on privacy-preserving recommender systems. For instance, differentially-private recommender systems (31, 29) ensure individual-anonymity, via careful injection of noise, while still enabling extraction of aggregate-level information used to feed the recommendation algorithm. In contrast, a consistency-based approach would hypothetically still allow for some individuals to be (more accurately) identified if they choose to be (e.g., to receive more targeted recommendations) while honoring the privacy of individuals who chose to not share their information.

**Towards industry reform.** Our results contribute to the growing sentiment that the digital advertising space is in need of reform, particularly due to the unintended consequences of "the collection of personal data, tracking, and massive-scale profiling" (2). As the move away from cookie-based tracking gains momentum, alternative mechanisms for user profiling are becoming more prevalent.

In the context of an advertising environment without cookies (often referred to as the *cookieless* future), the shift to these alternative data collection methods may lead to informational disparities across companies. Given that the current advertising ecosystem contains players of vastly different sizes, ending the use of cookies may give asymmetric power to companies that have already collected significant amounts of personal data. Indeed, tools that facilitate access to user information that would otherwise be tracked by cookies are already in use, e.g., websites that allow users to create an account by using their Google/Microsoft credentials (a practice known as "auto-linking").

In general, these issues point to the fact that controls and regulations should be based on the downstream effects of user interactions with the advertising system. In particular, policy must carefully consider how much information has been extracted from a given user, whether it be from individual decisions/responses of the user or gleaned from inferred similarities with other users, and its privacy-compromising effects. To foster transparency, it would be beneficial for the industry to derive estimates of the monetary value of user information and share them with the public. In general, the advertising industry can be more transparent, not just about user data but also concerning the downstream effects on users, since real-world advertising engines are vastly more complex than our simplified setting and may not lend themselves to simple mathematical modeling.

**Limitations.** Our findings on the disparate impact of consent should be interpreted with the understanding that our model necessarily has some limitations. Broadly, our model uses a simplified definition of cookies in which cookies serves as a proxy for the users' cohorts. While the reason for this simplicity is to extract insights that depend directly on the user's cookie consent decisions, extending the definition of a cookie to more realistic settings by including user click/behavioral information would likely generate additional insights. Secondly, to capture the core aspects of recommender systems, our recommendation model is based on foundational recommendation algorithms (namely online + confidence-weighted matrix factorization). While these algorithms form the basis for modern recommender systems, it would be worthwhile to see how the insights extend to more modern algorithms. Lastly, we consider simplified ad and user pools; consideration of additional ad features and a more realistic dynamic user pool could influence the findings.

# 7  Concluding remarks and future directions

We've investigated the question of how heterogeneous cookie consent rates among users influence the recommender agent's ability to learn users' latent features (e.g., their demographics and tastes). We've empirically discovered, through construction of a simulator, that disparities in the agent's estimation accuracy across users emerge when consent rates exhibit demographic-dependence. Our observations show that seemingly simple informational decisions by users (i.e., whether to share their cookie) can have complex effects on the agent's information. Consideration of informational effects of users' decisions is crucial in the design of recommender systems. We encourage the development of new fairness metrics for dynamic settings that enforce consistency between a user's privacy decision and the amount of information that the system knows about them.

Future work focuses on studying the downstream effects of our observations via validation in real-world recommender systems. An additional direction is quantification of the aforementioned fairness consistency property such that it can be embedded as a constraint in the design of recommender system algorithms.

**Acknowledgments.** This research was funded in part by the Horizon Europe project AutoFair (grant agreement ID: #101070568).

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

## A  Omitted expressions

**Derivation of interim belief update**. Given user consent decisions, the agent updates the prior $\mu$ to form interim beliefs as follows. For a user $i$ who provided consent ($x_i = 1$), with its revealed cookie $\phi_i = \varphi$, the interim belief $\tilde{\mu}_{i,0} \in \Delta(\Theta)$ is given elementwise by

$$
\begin{aligned}
\tilde{\mu}_{i,0}(\vartheta) &= \mathbb{P}(\theta_i = \vartheta \mid x_i = 1, \phi_i = \varphi) \\
&= \frac{\mathbb{P}(x_i = 1 \mid \phi_i = \varphi, \theta_i = \vartheta)\mathbb{P}(\phi_i = \varphi, \theta_i = \vartheta)}{\mathbb{P}(x_i = 1, \phi_i = \varphi)} \\
&= \frac{P(x_i = 1 \mid \theta_i = \vartheta)p(\phi_i = \varphi, \theta_i = \vartheta)}{\sum_{\vartheta'} \mathbb{P}(x_i = 1 \mid \theta_i = \vartheta')\mathbb{P}(\phi_i = \varphi, \theta_i = \vartheta')} \\
&= \frac{q_\vartheta \mu(\varphi, \vartheta)}{\sum_{\vartheta'} q_{\vartheta'} \mu(\varphi, \vartheta')}.
\end{aligned}
$$

Similarly, for a user $i$ who did not provide consent ($x_i = 0$), its cookie is not revealed and thus interim beliefs are given by

$$
\begin{aligned}
\tilde{\mu}_{i,0}(\vartheta) &= \mathbb{P}(\theta_i = \vartheta \mid x_i = 0) \\
&= \frac{\mathbb{P}(x_i = 0 \mid \theta_i = \vartheta)\mathbb{P}(\theta_i = \vartheta)}{\mathbb{P}(x_i = 0)} \\
&= \frac{\mathbb{P}(x_i = 0 \mid \theta_i = \vartheta)\sum_\varphi \mathbb{P}(\phi_i = \varphi, \theta_i = \vartheta)}{\sum_{\vartheta'} \mathbb{P}(x_i = 0 \mid \theta_i = \vartheta')\sum_{\varphi'} \mathbb{P}(\phi_i = \varphi', \theta_i = \vartheta')} \\
&= \frac{(1 - q_\vartheta)\sum_\varphi \mu(\varphi, \vartheta)}{\sum_{\vartheta'}(1 - q_{\vartheta'})\sum_{\varphi'} \mu(\varphi', \vartheta')}.
\end{aligned}
$$

**Cohort belief update.** Given a set of cohort beliefs across users $\mu = (\mu_1, \ldots, \mu_n)$ and a set of recommendation-response pairs $\mathcal{L}$, the agent forms updated beliefs $\mu' = (\mu'_1, \ldots, \mu'_n)$ via a Bayesian update, $\mu' = f(\mu, \mathcal{L})$. This update is carried out independently for each user $i$. Let $\mu_i$ denote the agent's current belief on user $i$' cohort $\theta_i$, and denote $\mathcal{L}_i = \{(a_i, c_{i,a})\}$ as the pairs that correspond to user $i$. The updated belief $\mu'_i$ is given by $\mu'_i = (\mu'_i(\theta^1), \ldots, \mu'_i(\theta^d)) = f_i(\mu_i, \mathcal{L}_i)$ where each $\mu'_i(\vartheta)$ is given by

$$
\begin{aligned}
\mu'_i(\vartheta) &= \mathbb{P}(\theta_i = \vartheta \mid \mathcal{L}_i) \\
&= \frac{\prod_{(a_i, c_{i,a}) \in \mathcal{L}_i} \mathbb{P}(C_{i,a} = c_{i,a} \mid A_i = a_i, \theta_i = \vartheta)\mathbb{P}(\theta_i = \vartheta)}{\sum_{\vartheta'} \prod_{(a_i, c_{i,a}) \in \mathcal{L}_i} \mathbb{P}(C_{i,a} = c_{i,a} \mid A_i = a_i, \theta_i = \vartheta')\mathbb{P}(\theta_i = \vartheta')} \\
&= \frac{\prod_{(a_i, c_{i,a}) \in \mathcal{L}_i^1} \bar{p}_{i,a_i}(\vartheta) \prod_{(a_i, c_{i,a}) \in \mathcal{L}_i^0} (1 - \bar{p}_{i,a_i}(\vartheta))\mu_i(\vartheta)}{\sum_{\vartheta'} \prod_{(a_i, c_{i,a}) \in \mathcal{L}_i^1} \bar{p}_{i,a_i}(\vartheta') \prod_{(a_i, c_{i,a}) \in \mathcal{L}_i^0} (1 - \bar{p}_{i,a_i}(\vartheta'))\mu_i(\vartheta')}
\end{aligned}
$$

where $\mathcal{L}_i^1$ (resp. $\mathcal{L}_i^0$) are the responses where the user clicked (resp. did not click) on the recommendation and $\bar{p}_{i,a}(\vartheta)$ is the expected click probability given $\vartheta$ defined as

$$
\bar{p}_{i,a}(\vartheta) = \mathbb{E}_{\alpha_i \sim \text{LogNormal}(\rho_\vartheta, \sigma_\vartheta^2)}[p_{i,a}(\alpha_i)]
$$

where the click probability $p_{i,a}$ from (1) has been written as $p_{i,a}(\alpha_i)$ to make the dependence on user $i$'s topic affinities explicit.

**Confidence weights.** The confidence weights $\bar{w}_{i,a,t}$ in (2) are computed as the expected binomial probability for seeing the current response counts $r_{i,a,t}$ given the agent's current belief on user $i$'s cohort $\vartheta_i$, i.e., $\bar{w}_{i,a,t} = \mathbb{E}_{\vartheta_i \sim \mu_{i,t}}[p_t(I_{i,a,t}, r_{i,a,t}, \vartheta_i)]$. The binomial probability $p_t(I_{i,a,t}, r_{i,a,t}, \vartheta_i)$ is defined as

$$
p_t(I_{i,a,t}, r_{i,a,t}, \vartheta_i) = \binom{I_{i,a,t}}{r_{i,a,t}} \bar{p}_{i,a}(\vartheta_i)^{r_{i,a,t}}(1 - \bar{p}_{i,a}(\vartheta_i))^{I_{i,a,t} - r_{i,a,t}}
$$

where $I_{i,a,t}$ is the current impression count for user-ad pair $(i, a)$, $r_{i,a,t}$ is the current click click for $(i, a)$, and $\bar{p}_{i,a}(\vartheta_i) = \mathbb{E}_{\alpha_i \sim \text{LogNormal}(\rho_{\vartheta_i}, \sigma_{\vartheta_i}^2)}[p_{i,a}(\alpha_i)]$ is the expected click probability.

# B  Recommendation procedure

---

**Algorithm 1:** Recommendation procedure.

---

*Input parameters.* $(n, l, m, \Theta, \Phi, \mathcal{T}, \{\rho_\theta, \sigma_\theta\}, u_0, \{q_\theta\}, \mu, k, \lambda, \varepsilon, \mathcal{L}_0, T_b, T)$

*Initialize users.* For each user $i \in [n]$, sample:

- cookie-cohort pairs $(\phi_i, \theta_i) \sim \mu$
- topic affinities $\alpha_i \sim \text{LogNormal}(\rho_{\theta_i}, \sigma_{\theta_i}^2)$
- consent decision $x_i \sim \text{Bernoulli}(q_{\theta_i})$

*Form interim beliefs.* For each user $i \in [n]$:

    **if** $x_i = 1$ **then**                      `// user provided consent`

        $\tilde{\mu}_i(\vartheta) \leftarrow \frac{q_\vartheta \mu(\phi_i, \vartheta)}{\sum_{\vartheta'} q_{\vartheta'} \mu(\phi_i, \vartheta')}, \; \vartheta \in \Theta$

    **else**                                `// user withheld consent`

        $\tilde{\mu}_i(\vartheta) \leftarrow \frac{(1-q_\vartheta) \sum_\varphi \mu(\varphi, \vartheta)}{\sum_{\vartheta'} (1-q_{\vartheta'}) \sum_{\varphi'} \mu(\varphi', \vartheta')}, \; \vartheta \in \Theta$

    **end**

*Offline responses.* Form priors using $\mathcal{L}_0$: $\mu_{i,0} \leftarrow f(\tilde{\mu}_i, \mathcal{L}_0), \; i \in [n]$

*Online recommendations.*

**for** $t = 1, \ldots, T$ **do**

    Define current ad pool $\mathcal{A}_t$ by sampling $l$ items uniformly without replacement from $\mathcal{T}$

    **if** $\;\text{mod}\,(t, T_b) = 0$ **then**                    `// retraining step`

        Update cohort beliefs: $\mu_{i,t} \leftarrow f(\mu_{i,t-1}, \mathcal{L}_t \setminus \mathcal{L}_{t-T_b}), \; i \in [n]$

        Compute weights: $\bar{w}_{i,a,t} \leftarrow \mathbb{E}_{\vartheta_i \sim \mu_{i,t}}[p_t(r_{i,a,t}, \vartheta_i)], \; i \in [n]$

        Update factor estimates:

$$(\hat{\boldsymbol{u}}_t, \hat{\boldsymbol{v}}_t) \leftarrow \operatorname{argmin}_{(u,v) \in \mathcal{U} \times \mathcal{V}} \left[ \sum_{(a_i, c_{i,a}) \in \mathcal{L}_t} \bar{w}_{i,a,t}(u_i^\top v_a - r_{i,a,t})^2 + \right.$$

$$\left. \lambda \left( \sum_{i \in [n]} \|u_i\|_2 + \sum_{a \in [m]} \|v_a\|_2 \right) \right]$$

    **else**

        Propagate cohort beliefs: $\mu_t \leftarrow \mu_{t-1}$

        Propagate factor estimates: $(\hat{\boldsymbol{u}}_t, \hat{\boldsymbol{v}}_t) \leftarrow (\hat{\boldsymbol{u}}_{t-1}, \hat{\boldsymbol{v}}_{t-1})$

    **end**

    Recommend ads: for each $i \in [n]$, recommend at $a_{i,t}$ via

$$a_{i,t} = \begin{cases} \operatorname{argmax}_{a \in \mathcal{A}_t} \hat{u}_{i,t}^\top \hat{v}_{a,t} & \text{w.p. } 1 - \varepsilon \\ a \sim \mathrm{U}(\mathcal{A}_t) & \text{w.p. } \varepsilon \end{cases}$$

    Append responses: $\mathcal{L}_{t+1} \leftarrow \mathcal{L}_t \cup \{a_{i,t}, c_{a,i,t}\}$

**end**

---

# C   Simulator and experiments

Our simulator was built upon `RecSim` (21) (source code at `https://github.com/emiehling/cookie-consent/`). The high-level architecture of our simulator is illustrated in Fig. 1 of Section 4. Additional details (with references to objects in the source code) are provided below.

**Advertisement and user samplers.** The advertisement sampler object (`AdvertisementSampler`) defines the distribution of each ad feature, here assumed to simply be the ad's topic. Similarly, the user sampler object (`UserStateSampler`) defines the distribution of each user feature, described by the joint cookie-cohort prior, the opt-in distribution, and the statistics of the user's topic affinities. The ad sampler and user sampler objects are used to define a gym environment for the recommendation procedure (via `MultiUserEnvironment` and `RecSimGymEnv`). Ads are resampled in each round whereas users remain fixed for the duration of the episode.

**Users and the recommender agent.** Each user is described by the class `RSUserModel`. This class contains the user's choice model, i.e., the logit model dictating the binary click decision given the recommended ad (see the method `simulate_response`).

Given the collections of ads and users, the recommender agent makes recommendations according to an $\varepsilon$-greedy bandit (see the pseudocode in Section B).

Retraining consists of first updating the cohort beliefs (via the methods `update_cohort_beliefs` and `get_click_probabilities`, see Appendix A for the expressions), updating weights $\bar{w}_{i,a,t}$, and recomputing the matrix factor estimates (via `get_estimated_factors`, see (2)). Estimation is carried out via stochastic gradient descent with a learning rate of $0.01$, regularization weight of $0.01$, and a stopping threshold on the mean-squared error of $\varepsilon_{\text{thresh}} = 0.001$. Latent factors are assumed to be of dimension $k = 50$.

**Experimental setup.** Simulations were run in Python 3.8 on an Intel(R) Xeon(R) CPU E5-2667 v2 (3.30GHz). Unless otherwise stated, baseline parameters of the simulation environment were as follows: $n = 1000$ users, $m = 200$ ads, ad candidate size $l = 50$, batch size $T_b = 1$, offline response set $\mathcal{L}_0 = \varnothing$, exploration probability $\varepsilon = 0.1$, and binary cookie and cohort spaces. Simulations were averaged over 500 runs/episodes.

The experimental setup can be extended in a variety of directions to investigate additional interesting questions. One direction is to extend the feature description of the ads (beyond topic) to include features that reflect ad quality and location. This would enable studying how the recommender system treats minority populations (compared to majority populations). Additionally, augmenting the simulator with the ability to handle a changing user pool would allow for analysis of the cold start problem.

# D Sensitivity analyses

Sensitivity analyses on cohort errors were carried out by independently varying the prior $\mu$, batch size $T_b$, offline response set $\mathcal{L}_0$, and topic affinity means $(\rho_0, \rho_1)$ with fixed parameters $n = 200$ users, $m = 200$ topics, ad pool size $l = 50$, latent factor dimension $k = 50$, and $|\Theta| = |\Phi| = 2$. Plots illustrate error means and standard deviations (red: $\theta = 0$, blue $\theta = 1$, solid: consent group, dashed: non-consent group) over 50 episodes. Baseline parameters are:

$$\mu = \begin{bmatrix} 0.4 & 0.1 \\ 0.1 & 0.4 \end{bmatrix}, \qquad T_b = 1, \qquad \mathcal{L}_0 = \varnothing, \qquad (\rho_0/m, \rho_1/m) = (0.3, 0.7)$$

**Prior.** The impact of the prior $\mu$ was studied in three cases dictated by the degree of informativeness of the cookie for inferring the cohort.

Partially informative: $\mu = \begin{bmatrix} 0.4 & 0.1 \\ 0.1 & 0.4 \end{bmatrix}$

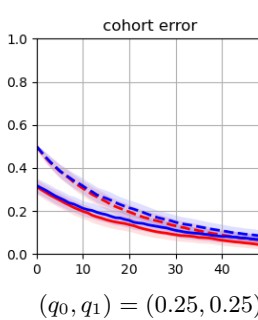 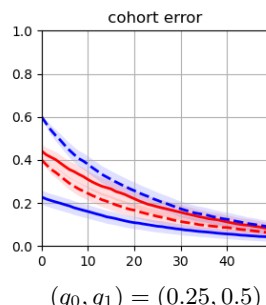 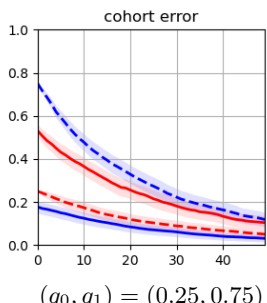

$(q_0, q_1) = (0.25, 0.25)$      $(q_0, q_1) = (0.25, 0.5)$      $(q_0, q_1) = (0.25, 0.75)$

Fully informative: $\mu = \begin{bmatrix} 0.5 & 0 \\ 0 & 0.5 \end{bmatrix}$

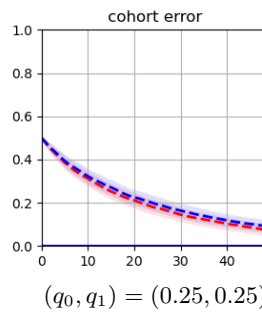 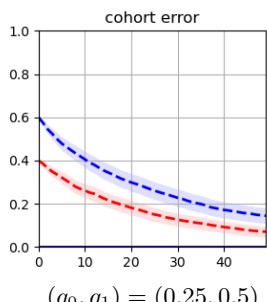 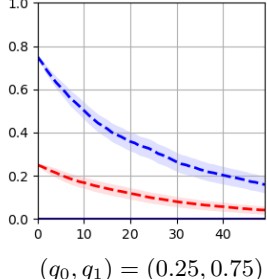

$(q_0, q_1) = (0.25, 0.25)$      $(q_0, q_1) = (0.25, 0.5)$      $(q_0, q_1) = (0.25, 0.75)$

Uninformative: $\mu = \begin{bmatrix} 0.25 & 0.25 \\ 0.25 & 0.25 \end{bmatrix}$

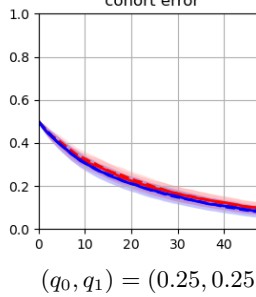 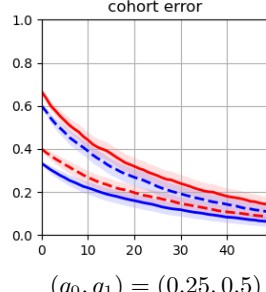 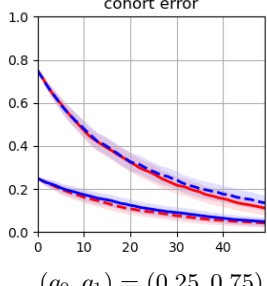

$(q_0, q_1) = (0.25, 0.25)$      $(q_0, q_1) = (0.25, 0.5)$      $(q_0, q_1) = (0.25, 0.75)$

For a partially informative prior (any $\mu$ that does not have identical rows), knowledge of a user's cookie partially reveals the user's cohort. The resulting cohort errors are consistent with Fig. 2 of Section 5.1. For the fully informative prior, the agent is completely certain of users' cohorts for users who opted-in (the agree group), but still possesses uncertainty for users who did not opt-in (the disagree group). Lastly, for the uninformative prior, revelation of a user's cookie does not inform the user's cohort (as the likelihoods of seeing cookie values are identical across cohorts) and the agent must infer cohorts solely from differences in response behavior.

**Batch size.** The batch size, $T_b$, dictates how many responses to collect from each user before retraining. Values for the batch size were varied in the range $T_b \in \{1, 2, 10\}$.

$\underline{T_b = 1}$:

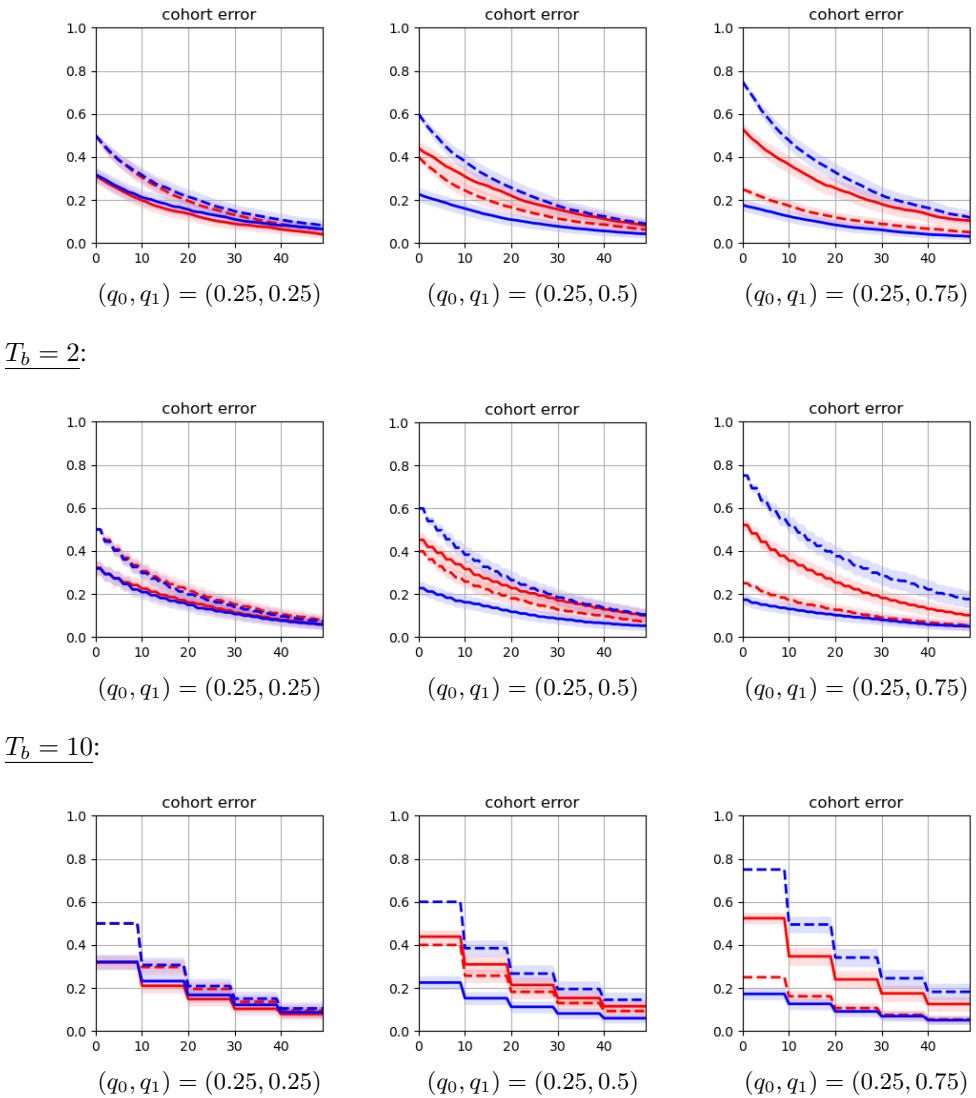

Intuitively, for the same number of observations, cohort estimation errors are the same across various batch sizes (e.g., updates may be less frequent but they contain more data). However, waiting until a batch update ($T_b > 1$) results in more interactions where the users face greater disparate estimation errors (compared to the $T_b = 1$ case).

**Size of offline response set.** The offline response set, $\mathcal{L}_0$, is a set of recommendation-responses that are available before the online recommendation process. Recommendations in the offline set, $\mathcal{L}_0$, were generated uniformly at random with responses generated by the users' choice models. Simulations were run for $|\mathcal{L}_0| = \{0, 1, 5\}$, differing in the number of offline responses assumed available from each user.

$|\mathcal{L}_0| = 0$:

$|\mathcal{L}_0| = 1$:

$|\mathcal{L}_0| = 5$:

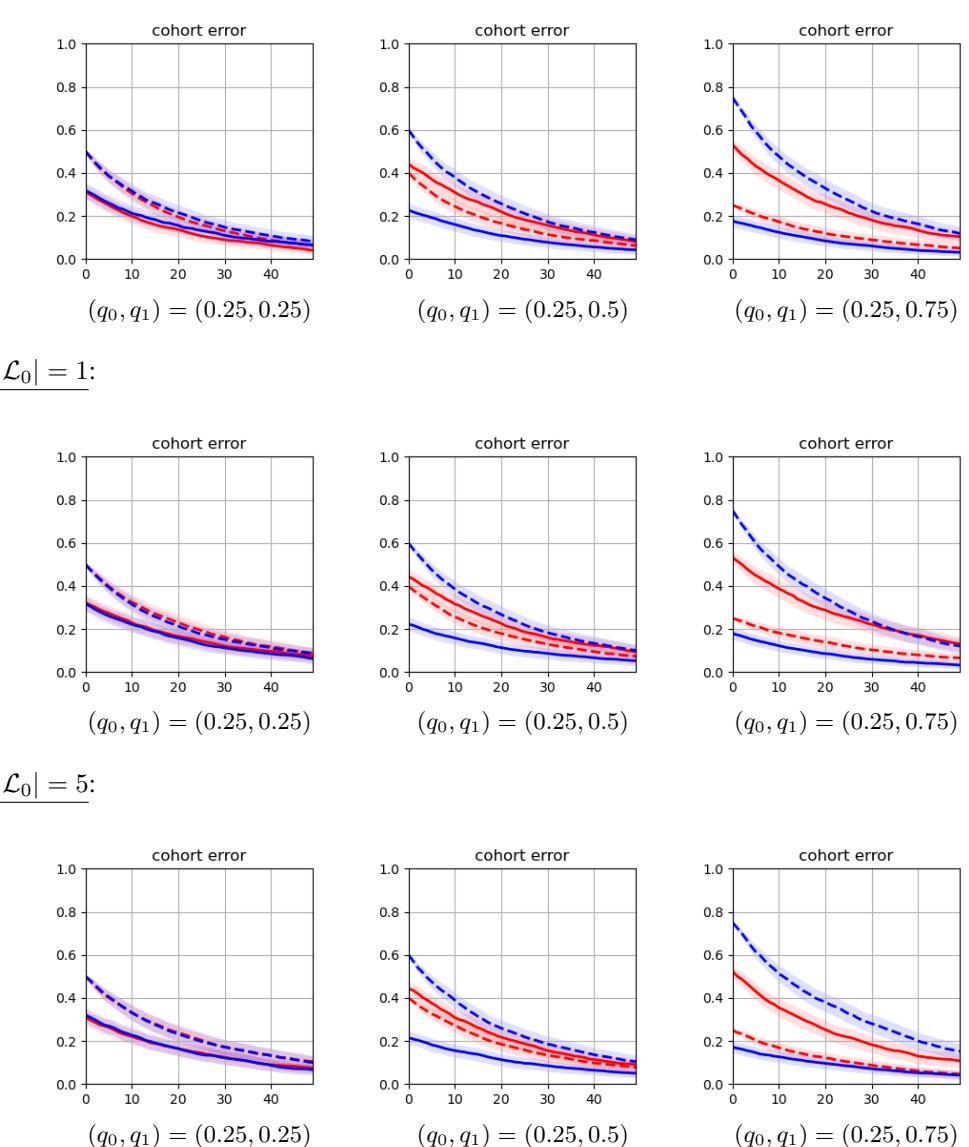

The impact of the offline response set does not appear to have a significant effect on the evolution of the cohort errors.

**Similarity of topic affinities.** The sensitivity to similarity of topic affinities was studied by increasing similarities of the synthetic affinities via means (normalized by topic count $m$) $(\rho_0/m, \rho_1/m) \in \{(0.3, 0.7), (0.4, 0.6), (0.5, 0.5)\}$.

$(\rho_0/m, \rho_1/m) = (0.3, 0.7)$:

$(\rho_0/m, \rho_1/m) = (0.4, 0.6)$:

$(\rho_0/m, \rho_1/m) = (0.5, 0.5)$:

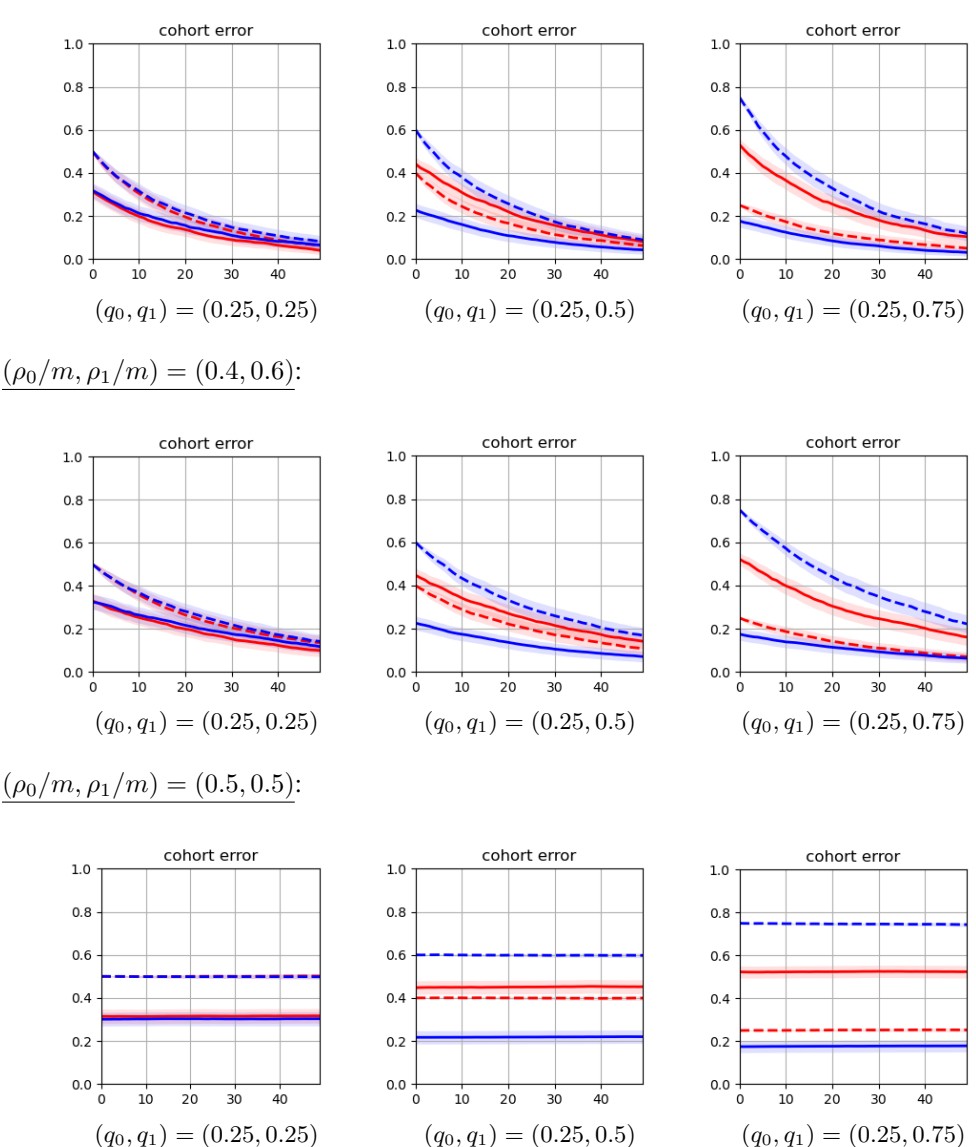

The agent's ability to distinguish users based on their responses depends on the similarities of affinities across users in different cohorts. Intuitively, as the topic affinities across cohorts become more similar, the agent requires more responses to reach the same level of estimation error (since users from different cohorts behave more similarity as topic similarity grows). The extreme case of identical statistics of users' affinities across cohorts $((\rho_0/m, \rho_1/m) = (0.5, 0.5))$ results in the agent being unable to resolve any uncertainty over users' cohorts (since user responses are uninformative for their cohort).

