# OpenReview forum: "Cookie Consent Has Disparate Impact on Estimation Accuracy"
_NeurIPS.cc/2023/Conference — NeurIPS 2023 poster_

### Official Review · Reviewer_WBi3 · 2023-06-29

**Soundness:** 2 fair
**Presentation:** 3 good
**Contribution:** 2 fair
**Rating:** 4
**Confidence:** 4

**Summary:**

Cookies enable accurate identification and tracking of user behavior, leading to personalized ads and better ad campaign performance. However, this raises privacy and fairness concerns. This study investigates the impact of user consent on a recommender system's ability to learn about users' demographics and preferences. It reveals that when consent rates are demographically dependent, a user's choice to not share their cookie can paradoxically lead to the recommender system knowing more about them. Moreover, the gap in consent rates between demographics amplifies estimation errors. As the system receives more user responses, the effects of consent decisions diminish. The findings highlight the need for new fairness notions that promote consistency between users' privacy choices and the accuracy of the recommender system's estimations.

**Strengths:**

The paper studied a practical topic which is valuable to the community. The paper is also clearly written and easy to follow.

**Weaknesses:**

1. The work seems incomplete. With a simulation experiment and an analysis, there's no solution to the observed issues.
2. The simulation setting is a bit simplified, introducing inductive bias to the results. Concretely, matrix factorization only mimics user behaviors (e.g., click), ignoring the user/item features. To be honest, matrix factorization is not the mainstream technique in industry. It is tree model and deep model. These model will be affected by user consent decision since user features will be unavailable, which will pose challenges to model training and serving, and then model performance and subsequent fairness, privacy issues. It will be more attractive if authors consider this setting.


**Questions:**

1. User grouping/clustering is a common practice in industry. Estimating effects of cookie consent stuff with real-world dataset is more meaningful. I would suggest analyze performance and consent rate with the external grouping info

**Limitations:**

N/A since the authors didn't propose new methods.

---

> ### Author Rebuttal · Authors · 2023-08-09
>
> Thanks for the review.
>
> - Regarding your comment that our paper does not contain a solution, our view is that there is value in drawing attention to an issue without necessary having a fix. We’ve proposed changes to industry practice that should be made, along the lines of ensuring that how much a system knows about a user is consistent with their privacy decision, but we feel that by no means is the solution trivial. It will likely require deliberate loss of user information, which in turn will require the design of incentives to do so. The goal of our paper is to present an interesting observation related to the fairness of consent with the hopes of seeding further conversation on the topic.
>
> - In response to your comment on the simplicity of the simulation setting, we agree that it would be interesting to investigate the impact of user consent in the context of more realistic recommender system algorithms, however, we would like to note that (while a simple algorithm) matrix factorization does form the basis for many more modern recommender algorithms (hybrid and embedding-based methods). The goal of our paper was to illustrate the disparate impact of consent for a simple foundational algorithm such that these insights would encourage broader discussion in the community (and, to your point, investigation into if these effects also exist in more complex systems). Additional discussion on the simulation setting can be found in the global rebuttal section under “Real-world relevance of the studied model.”
>
> - Lastly, grouping or clustering of users is done in practice to aid with scalability — clustering reduces the effective dimension of the recommendation task by making group-level recommendations (generally at the cost of recommendation accuracy). While this is an important algorithmic step for practical recommender systems, the focus of our paper is on the impact of user-level consent decisions. Our understanding of your comment is to extend this study to include alternative grouping structures (not necessarily dictated by the user’s cohort), which is certainly an interesting direction, but we feel falls outside the main message of our paper.

---

> > ### Comment · Area_Chair_dRtb · 2023-08-21
> >
> > I'd like to thank the authors for their clarification and response to some of the reviewers' major concerns.

---

### Official Review · Reviewer_S9vB · 2023-07-06

**Soundness:** 2 fair
**Presentation:** 2 fair
**Contribution:** 1 poor
**Rating:** 4
**Confidence:** 4

**Summary:**

* This paper aims to investigate the effect of cookie consent (as mandated by GDPR, ePD), on the accuracy of recommendation algorithms using a simulation study.
* Model:
  * Each advertisement $a$ is characterized by a topic $\tau \in \mathcal{T} \cong [m]$.
  * Each user $i\in[n]$ described by a cookie $\phi_i \in \Phi \cong [c]$, a demographic $\theta_i \in \Theta \cong [d]$, and vector of topic affinities $\alpha_i \in \mathbb{R}_+^m$.
  * Prior $\mu\in\Delta(\Phi\times\Theta)$ is assumed over cohort-cookie pairs. Users from cohort $\theta$ reveal their cookie with independent probability $q_\theta$. User preferences $\alpha_i$ are multivariate log-normal.
Probability of user $i$ clicking ad $a$ (denoted $c_{i,a}=1$) is $p_{i,\tau}$, depending on the the utility and on a “no-click-mass” parameter $p_0$.
  * Upon observing the user's cookie (or lack of consent), the system constructs a posterior belief on their cohort $\bar{\mu}{i,0}\in\Delta(\Theta)$. Items are recommended by an $\varepsilon$-greedy policy, and the model is retrained at each time step $t$ using regularized least squares.
* In the main experiment, the authors simulate the environment using the Recsim framework, for a population with two cohorts, two types of cookies, $n=200$ users, and $m=200$ topics.
  * Section 5.1 demonstrates disparate impact of consent - Errors are symmetric in the case of homogeneous consent, and disparate in the case of heterogeneous consent.
  * Section 5.2 aims to demonstrate amplification effects, showing the gap in consent rates serves as an amplifier for an individual’s consent decision.
* Finally, the authors discuss in detail the possible implications of such effects on popular notions of fairness, and the effect of cookie of consent on the industry.


**Strengths:**

* Problem is novel and well-motivated.
* Model is outlined clearly.
* Documented code is provided, and sensitivity analysis was conducted on the provided model.


**Weaknesses:**

* Paper only analyzes the behavior of a specific, non-standard recommender in a fully-synthetic environment. It is not clear from the analysis whether such effects are significant compared to other factors which are currently ignored in practice.
* Disparity between the consent group vanishes as $t$ grows, raising concerns about the significance of the claimed effects in practice
* Experiments compare relatively simple populations: Only two populations, with relatively low dataset sizes (fixed pool of $n=200$ users). Such experiments can be complemented either by theoretical analysis or experiments on real-world data, but neither are provided in the paper.
* Minor technical remark: The letter $a$ is used to denote both ads (e.g in section 4), and “agreement to collect cookies” (e.g in section 5.2), which caused confusion. The notation $\bar{\mu}$ is not referred to in the algorithm.


**Questions:**

* What is the effect of the recommendation algorithm on such accuracy disparity effects? Do recommendation algorithms that are indifferent to cookie consent amplify or diminish disparity? (e.g matrix factorization with uniform weights instead of uniform, KNN, etc.)
* What is the motivation behind the modeling of cookies as a low-dimensional categorical variable ($\phi \in \Phi \cong [c]$)? How does it reflect the common structure of tracking cookies in practice? As far as I understand, cookies contain unique identifiers that track users across visits, and are therefore unique to every user. Moreover, users can store multiple types of cookies, and not just one.
* How does the model behave when there are more than 2 cohorts? How does it behave when more than 2 categorical cookie types?
* In which ways does the synthetic simulation differ significantly from practical settings? Do we expect the magnitude of disparity effects to increase or diminish when discussion moves to practical settings? Why, and is it possible to support the claim with experiments?
* Can similar disparity effects be derived from simpler principles? (i.e the “Missing at Random” model from survey statistics causing bias when unaccounted for?)


**Limitations:**

I feel that the authors did not discuss limitations to a sufficient extent. Claimed effects are only demonstrated in a synthetic simulation with a non-standard recommendation algorithm. Even though I strongly agree that discrepancy effects may occur in practice to some extent, the analysis does not indicate whether their magnitude is significant in practical settings.

---

> ### Author Rebuttal · Authors · 2023-08-09
>
> We appreciate your careful review. We hope that our responses below resolve some of your concerns.
>
> Impact of recommendation algorithm on accuracy disparity:
>
> - The impact of different recommendation algorithms on the observed disparity differs depending on how the algorithm deals with missing information. Generally, algorithms that are indifferent to consent would treat all users identically (ignoring the cookie information of consenting users). Matrix factorization with uniform weights is one such algorithm — all user-item interactions would be weighed the same likely leading to lower disparity. For KNN, the situation is more complex, where depending on how many “neighbors” a user has (dictated by the consent decision) it could be possible that disparity would be amplified as the measure of similarity becomes less reliable. In general, whether an algorithm diminishes or amplifies disparities depends on how it deals with the user’s missing cookie information (in the case of non-consent).
>
> Modeling of cookies:
>
> - We agree that our cookie model is simplified. In practice, cookies generally contain both static information (e.g., user’s region) that is informative for their demographic/cohort as well as dynamic/behavioral information (e.g., click and browsing behavior). As part of our initial investigation, we’ve decided to focus on just the static component.
>
> - While we haven’t included behavioral information into a user’s cookie for the purposes of this paper, this is certainly an interested direction. As an initial thought, given the essentially unique cookie value for each user, revealed cookies would provide a significant amount of information on the identity of the user, potentially increasing the gap in treatment between consenting and non-consenting users. This is certainly a topic that we would like to investigate in follow-up work.
>
>
> Model behavior for more than 2 cohorts/cookies:
>
> - We’ve run additional simulations in response to your question (see the attached pdf). Explanation of the results for an increased number of cookies and cohorts can be found in the global rebuttal (under “Cookie and cohort model”) as it was raised by another reviewer.
>
>
> Differences between simulation and practical settings:
>
> - Our model differs from practical settings in multiple ways. As discussed in the global rebuttal section, our model is intentionally simplified in order to isolate the effects of cohort-dependent consent rates without the noise or confounding factors present in practical settings (factors that would mask our ability to detect such disparities but may still exist). We do not know with certainty how the degree of disparity would change in practice (it’s possible that disparities may increase with dynamic information as discussed above). We argue that the findings from our simplified setting should at least provide some motivation for investigating if these issues also exist in practice.
>
>
> Disparity effects from simpler principles:
>
> - Your question regarding the “Missing at Random” model is interesting, however, we feel that this is a different setting from our cohort-dependent consent setting. The MAR model assumes that the probability that a value is missing is related to the observed data but is independent of the unobserved data. Mapping this to our consent setting, this would imply that a user’s consent decision is based solely on observed quantities, without any influence from the underlying unobserved quantities (e.g., cohort). Our model specifies that consent is dictated by the unobserved quantity (cohort), which differs from the MAR model.
>
> Other comments:
>
> - Regarding your comment on the disparity vanishing as the number of interactions grow, this occurs after a relatively high number of interactions, with the disparity present for a significant number of interactions (upwards of 20 interactions). We lack precise data on the average number of sequential interactions that a user makes with a given website (or an ad serving platform), however, there are many reasonable scenarios where a user will only interact with a website a handful of times before navigating elsewhere (not to mention users periodically clearing their cookies, restarting the interaction counter as a result).
>
> - While we did not run simulations on a dynamic user pool due to the time constraints for the rebuttal, we would imagine that the impact of a such a modification may further amplify the observed disparities. The reason for this is because the gap in estimation accuracy between consenting and non-consenting users is greatest when the interaction count is low; when the user pool is changing, there will be more fresh users in the user pool who, by definition, have zero interactions with the recommender system.
>
> - We've included a summary of the limitations of our paper in the global rebuttal section.

---

> > ### Comment · Reviewer_S9vB · 2023-08-18
> >
> > Thank you for the thorough response! It helped clarify, and the additional results you describe sound very interesting.
> >
> > In my perspective, the main claims of the paper seem to rely on two non-trivial elements: (1) a novel recommendation algorithm that differs significantly from algorithms commonly used in the literature and in practice, and (2) an interesting, but highly non-trivial model for internet cookies. As these two elements are evaluated together, the contribution of each element can't be isolated, making it more difficult to apply the results and build upon them.
> >
> > The extended simulations and results in the reply to reviewer cPvy suggest that effects persist even under more common recommendation algorithms, thereby isolating the effect of element (2) to some extent. However, despite the additional reasoning, the current cookie model still does not seem intuitive to me, and I believe it needs to be further justified. Additionally, while I regard the new results as essential to the key claims of the paper, they cannot be thoroughly validated. I also agree with the points raised by reviewer iC41. I therefore update my rating to 4 (Borderline Reject).

---

### Official Review · Reviewer_cPvy · 2023-07-07

**Soundness:** 4 excellent
**Presentation:** 4 excellent
**Contribution:** 3 good
**Rating:** 7
**Confidence:** 3

**Summary:**

This paper simulates a recommendation system that is aware of and responds to users decision to share cookies, along with data from users' clicks. In this simulation, the paper finds that the recommendation accuracy could be higher for users who do not consent to provide cookies under certain conditions. The paper empirically analyzes these conditions along various dimensions.

**Strengths:**

The paper is very well written, and the presented result is novel to the best of my knowledge. The paper has a high potential to spur future work on the interplay between cookie consent decisions and recommendation accuracy.

**Weaknesses:**

**Strong claim in the title**

The key finding is via simulations. The title makes it seem like the result of a randomized field experiment. I think it is possible to temper the title to reflect this.

**Complex simulation setup, results possibly driven by unrealistic recommendation model*

The simulation setup is not parsimonious, which makes it hard to see the key drivers of the result.

   a. Why is the ad pool sampled in each round and not deterministic?
   b. Why does the cookie contain demographic information, and not contain user click/navigation history information?
   c. The recommendation model is designed to be click-maximizing for the given user model. In practice, the recommendation system has no idea about the user model and likely just performs matrix factorization (with or without exploration) to generate recommendations. This modeling choice for the recommendation model is a large deviation from practice; what happens to the results if the recommendation model is just user model-agnostic matrix factorization?

**Cookie-cohort dependency and cookie model assumptions**

The cookie in the user model is designed to have the sole function of being a proxy for the user's cohort. Specifically, it is not "contaminated" by what the user actually clicks on, which deviates from reality (cookies track clicks and navigation history to some extent). Combined with the demographic disparity in consent rates, this turns the presence or absence of cookies (and the cookies themselves) into unrealistically strong correlates of the cohort of an individual.

How do the results change if the cookies also stored some sort of "average empirical topic preferences" over the users' last N clicks? This reduces their strength as a prior and makes them less useful, since they reflect data that the recommendation model has already seen. However, I think this is more realistic.

**Theoretical explanations for the empirical observations**

How do the observations deviate from theoretical predictions?

Specifically, Bayes' rule can be used to show that a demographic demographic disparity in consent rates will increase the accuracy of predicting a user's cohort from their consent decision. Further, the cohort with a higher consent rate will provide more data to estimate that cohort's topic preferences accurately in each round. These two factors combined will influence the overall recommendation accuracy until the recommendation system has collected enough clicks from both cohorts.

As such, Observation 1 and 2 seem to say that the outlier groups in each cohort (with respect to the consent decision) have a higher estimation error. Does Figure 3 contradict this explanation?

My low overall rating is primarily a function of the lack of theoretical explanations.

**Questions:**

See weaknesses above

**Limitations:**

The paper does not discuss any limitations. I think discussing the limitations is important, given the assumptions made in the simulation.

---

> ### Author Rebuttal · Authors · 2023-08-09
>
> We appreciate your detailed review of our paper. We’ve responded to each of your queries (either in this response or in the global rebuttal).
>
> Response to: "Complex simulation setup, results possibly driven by unrealistic recommendation model":
>
> - Practically, the set of recommendable ads may be fixed, (re)sampled, or generated by some other process (e.g., via some auction process); see a discussion in the simulation framework upon which our code is based [Ie et al., 2019]. We chose a resampled ad pool because we felt that it was more realistic than a fixed ad pool (real-world ads are continually coming and going). Additionally, from an algorithmic perspective, a resampled ad pool helps with inference/learning due increased diversity of ads across a given episode. We’ve included a plot in the one-page attachment to illustrate the effect of a deterministic/fixed ad pool.
>
> - Your comment regarding the nature of information contained in the cookie was raised by another reviewer as well. Please see the global rebuttal (under "Cookie and cohort model") for our response.
>
> - We would like to address your comment that “in practice, the recommender system has no idea about the user model.” While we agree that most recommender systems may not have access to an explicit user model, the system is not blind to user behavior/preferences. Real-world recommender systems base recommendations on learned user preferences (from vast amounts of user-item interaction data) in order to maximize the some measure of engagement (e.g., clicks, shares, watch time, etc.). These data-driven models serve as models of user behavior, even if implicit. Consequently, we argue that no recommendation model is completely model-agnostic, but we would be happy to discuss more on some model variants we can investigate.
>
>
> Response to: "Cookie-cohort dependency and cookie model assumptions":
>
> - We appreciate your comment regarding incorporating click behavior into the description of the user’s cookie. Under your proposed modification, a cookie would be represented as a pair (cookie value, empirical topic preferences). Given that both the consent rate and a user’s tastes are cohort-dependent, we feel that incorporation of average empirical topic preferences into a dynamic component of the cookie would still exhibit strong cohort-dependence (rather than be contaminated by user behavior). We’d like to incorporate this into the paper but given the time constraints of the rebuttal period we were unable to include this modification.
>
>
> Response to: "Theoretical explanations for the empirical observations":
>
> - Regarding your comment on theoretical explanations for the empirical observations, observation 1 states that, in the case of cohort-dependent consent rates, users who disagree (to sharing their cookie) experience a lower estimation error only if they belong to the lower consent rate population (the intuitive non-consent leading to higher errors only holds for the higher consent rate population). Note that this effect holds purely due to the difference in the consent rates between the two cohorts, and is not dictated by the relative size of the agree vs disagree groups in each cohort in general (in response to your statement on “outlier groups”). In other words, even in the case where both cohorts have more users that disagreed vs agreed (see the case where $q_0=0.1$ and $q_1=0.3$). Observation 2 explains how two users from different cohorts fare when making a given consent decision. These observations appear consistent with the figure.
>
> Other comments:
>
> - We’re currently discussing some modifications to the title without deviating too much from the main finding of our experiments (disparate impact of consent).
>
> - We've also included a summary of the limitations of our paper in the global rebuttal section.
>
> - Lastly, while we agree that our paper is empirical in nature, we feel that this brings sufficient value due to the potential to encourage additional conversations (and experimentation) in this space, even in the absence of theoretical explanations.

---

> > ### Comment · Reviewer_cPvy · 2023-08-12
> >
> > Thanks for the response! It mostly addresses my concerns, but let me start with one remaining concern.
> >
> > My earlier comment said that “in practice, the recommender system has no idea about the user model.” Your response says "the system is not blind to user behavior/preferences." However, this is not what I claimed. I claimed that *your specific user model* is inaccessible to the recommender system (i.e. the form of the model). I agree that revealed preferences are always available for recommender systems to use.
> >
> > More concretely, I would model a recommender system parsimoniously as a matrix factorization of the user-item interaction matrix. More complex recommenders are possible, but this one is a canonical textbook example. Given a recommender system that simply factorizes the user-item interaction matrix, do the results hold?

---

> > > ### Author Response · Authors · 2023-08-16
> > > **Impact of a simpler recommendation system**
> > >
> > > Thank you for clarifying. In response to your comment, we’ve just finished running some additional simulations with a recommendation agent that computes factor estimates via a standard matrix factorization algorithm (without heterogenous confidence weights, i.e., all responses are treated with equal importance). To ensure that the factorization is meaningful (i.e., there is something to factor), a set of 10 recommendation-response pairs (obtained uniformly and offline) are used for initial estimates. Updated factor estimates are obtained every 10 interactions. Unfortunately we’re unable to attach plots to this response but we’ll do our best to explain the main insights.
> > >
> > > The new simulations indicate that the disparity effects still persist even in the absence of confidence weights (namely observations 1 and 2 still hold). The reason for this is because even though under the standard MF procedure where the algorithm is not maintaining confidence weights on user responses, the user responses still carry the same information on user cohort’s as they do under the more complex (confidence-weighted) model. This is simply because for a given set of factor estimates, the (Bayesian) inference process is only a function of the cohort beliefs, the recommendations, and the associated user responses.
> > >
> > > Your comment is an important one, since it highlights that the disparity issue is not simply an artifact of the specific confidence-weight matrix factorization procedure used in the paper, but arises from the more fundamental interaction between cohort-dependent consent rates and the system’s ability to accurately infer user cohorts based on their responses.
> > >
> > > We intend to include this insight as part of a section in the supplemental material on the impact of various recommendation algorithms.

---

> > > > ### Comment · Reviewer_cPvy · 2023-08-19
> > > >
> > > > Thanks for the hard work! I know external links and images are not shareable (except via the area chair). In any case, these insights are very useful and I believe strengthen the paper. I also take back my earlier comment on "unrealistic recommendation model" in light of these insights, and have increased my score by one.

---

### Official Review · Reviewer_iC41 · 2023-07-24

**Soundness:** 3 good
**Presentation:** 3 good
**Contribution:** 2 fair
**Rating:** 4
**Confidence:** 3

**Summary:**

This paper investigates the influence of varying consent rates on the accuracy of estimation. The paper utilizes a simulation-based approach, in which the decision to share cookies plays a significant role.
The findings reveal interesting patterns that can contribute to the development of more refined fairness metrics, taking into account the individual's choice to share information.


**Strengths:**

1.The paper presents an intriguing explanation of the observed phenomena in the simulation experiment, particularly the behavior of low-agree individuals (such as older people) choosing not to share information, which instead results in a lower estimation error for the agent in a cold-start setting.

2.The author's discuss on how these results can guide future directions to enhance the consistency of fairness evaluations.


**Weaknesses:**

1.The study's conclusions are based solely on a simple simulation experiment. There is a lack of discussion or testing on whether these observations would hold or approximately hold in real-world datasets. This limits the generalizability of the findings and their applicability to practical scenarios. Please correct me if I missed the details.

2.The observations (1 and 2) seem like they could be derived from trivial calculations using the prior $\mu_{i,0}^{-}$ without the need for simulation. (See Questions)


**Questions:**

Q1: Can the observations in Fig.2 at t=0 be simply demonstrated by your calculated prior $\mu_{i,0}^{-}$? Is there a significant difference between the priors $\mu_{i,0}^{-}$ without the recommendation simulation and $\mu_{i,0}$ with the offline responses?

Q2: The demographics and cookies are set to a specific number of 2. If the attribute number is changed (e.g., to 3, 4, etc.), do the observations still hold?

Q3: What is the trend of cohort-dependent consent rates on cohort beliefs $\mu$, and weights W? Could you provide a similar illustration as in Fig.2 for estimation errors?


**Limitations:**

See weaknesses and questions

---

> ### Author Rebuttal · Authors · 2023-08-09
>
> Thank you for your review. We've provided some answers to your questions below.
>
> The need for simulations:
>
> - While the priors do indeed dictate the relative accuracy disparities across cohorts and user consent decisions, the evolution of these errors as a consequence of the recommendation process is not straightforward. We’ve run some additional simulations to emphasize this point. In particular, see the plots in the 1-page attachment studying the resulting CTRs (under the caption “Impact on recommendation quality”) as well as how the cohort belief errors behave for an increased number of cookies/cohorts ("Effect of model size parameters").
>
> - Generally, there does seem to be an impact of consent on recommendation quality, however, the specific nature of the impact is not clear. We were originally planning on including these results in the initial submission, however, given that we lack a clear “observation” of the impact of consent on recommendation quality, we decided to instead present our findings on the impact of consent on estimation accuracy (we can however include these in the supplemental material for completeness). We feel that the impact on accuracy is a sufficiently interesting observation to encourage the community to investigate this issue further (in both simulated and real-world settings).
>
> Impact of number of cookies/cohorts:
>
> - Our additional simulations in the attachment illustrate the behavior of the model under both $c=d=3$ and $c=d=4$
>
> - The primary observations still hold as the number of cookies and cohorts increases. See the section "Cookie and cohort model" in the global response section for a more detailed description (as this query was raised by another reviewer as well).
>
> Relationship between consent rates and cohort beliefs and weights:
>
> - The impact of consent rates on cohort beliefs can be derived directly from the Bayes update equations. Generally, higher consent rates for a particular cohort will increase the belief (higher confidence) that users are from that cohort if they opt-in, with lower consent rates having the opposite effect. Similarly, given that the weights are the expected binomial probability of the current response counts, a higher consent probability will lead to a higher weight (reflective of a higher confidence for the given counts).

---

> > ### Comment · Reviewer_iC41 · 2023-08-18
> > **Thanks for your response**
> >
> > Thank you for your additional information.
> > I agree with the author's explanation that, although the simulated scenario may differ from the real-world scenario, it can eliminate the interference of noise and allow us to observe some potentially useful conclusions. For this, I can hold a reserved opinion. However, in other aspects, I still believe that the reasons to reject this paper outweigh the reasons to accept it, and I will keep my score unchanged.
> > Below are the reasons I have considered:
> >
> > Reason to accept: If this work is intended to encourage the community to further investigate these new questions, perhaps accepting this paper is justifiable.
> >
> > Reason to reject: However, as a work that proactively proposes new questions and observes phenomena, in my opinion, the observations made in this paper do not provide sufficient inspiration, which diminishes the paper’s appeal:
> >
> > Although observations 1 and 2 seem interesting because they show differences, as the authors also acknowledge, they can actually be simply predicted based on the setting of priors without simulation.
> > Observation 3 is a conclusion truly obtained through simulation. The results related to Observation 3 in the paper (where the differences in observations 1 and 2 vanish), as well as the impact on recommendation quality outlined in the newly provided appendix by the author, seem to indicate that whether cookies are considered or not does not have a significant effect on subsequent recommendation success or on the belief of the cohort. This could undermine the motivation to further investigate this question.

---

### Author Rebuttal · Authors · 2023-08-09

Real-world relevance of the studied model:

- We appreciate the reviewers’ concerns regarding the relevance of the model to real-world settings. We’d like the emphasize that the goal of this investigation is to present an intentionally simplified model in order to isolate and understand the effects of specific model aspects (i.e., impact of different user consent rates) without the additional noise and confounding factors present in real-world settings.

- That being said, we were careful when designing our model to ensure that it captured the core aspects of real-world recommender systems, namely confidence weights [Hu et al., 2008; Koren et al., 2009] and continual/online update of factor estimates via online methods [Wang et al., 2017]. These seminal papers form the foundation of many (more complex) models deployed in practice. As such, a primary takeaway of our paper is that there is at least a cause for concern that our findings may also exist in production-level recommender systems that are based on these fundamental algorithms. We will include a segment on this discussion in the revised paper.

- We feel that this philosophy (of gaining insights for a simplified version of a model) is a powerful approach to research in complex AI systems, as many negative effects can be hidden in the complexity of real-world models, but still present under the surface.


Cookie and cohort model:

- A couple of reviewers have raised questions on the choice of representing cookies as static quantities. We agree that in reality cookies contain both static information (e.g., location information) and dynamic information (e.g., user click and browsing behavior). Our decision to model user cookies as only containing static information (as a proxy for user cohort) was an intentional simplification to isolate the effects of cohort-dependent consent rates (which users are more or less likely to provide their consent) on the recommendation process and investigate the potential fairness issues on specific user groups. In practice, the user profile (as opposed to the actual cookie value) is what drives recommendations — we abstract this by representing cookies as proxies for the user’s core identity (cohorts/demographics).

- Regarding the impact of a larger number of cookies/cohorts, we’ve run some additional simulations under two cases: i) $c=d=3$, and ii) $c=d=4$, each under a variety of cohort-dependent consent rates. Please see the attached pdf under the caption “Effect of model size parameters.” As seen from the plots, the primary observation still holds, with a necessary modification: non-consenting users in any cohort that is not the maximal consent-rate cohort experience lower estimation errors for disagreeing (non-consent) than agreeing (consent). We will augment the observations in the paper to include this generalization.


In response to the reviewer’s comments, we’ve consolidated the limitations of our approach (to be included as a limitations section of the revised paper):

- Our findings on the disparate impact of consent should be interpreted with the understanding that our model necessarily has some limitations. Broadly, our model uses a simplified definition of cookies in which cookies serves as a proxy for the users’ cohorts. While the reason for this simplicity is to extract insights that depend directly on the user’s cookie consent decisions, extending the definition of a cookie to more realistic settings by including user click/behavioral information would likely generate additional insights. Secondly, to capture the core aspects of recommender systems, our recommendation model is based on foundational recommendation algorithms (namely online + confidence-weighted matrix factorization). While these algorithms form the basis for modern recommender systems, it would be worthwhile to see how the insights extend to more general algorithms. Lastly, we consider a fixed user pool; consideration of a more realistic dynamic user pool could influence the findings.

References (for both global rebuttal and individual rebuttals):

- [Hu et al., 2008] Yifan Hu, Yehuda Koren, and Chris Volinsky. "Collaborative filtering for implicit feedback datasets." In 2008 Eighth IEEE international conference on data mining, pp. 263-272. IEEE, 2008.

- [Ie et al., 2019] Eugene Ie, Chih-wei Hsu, Martin Mladenov, Vihan Jain, Sanmit Narvekar, Jing Wang, Rui Wu, and Craig Boutilier. "Recsim: A configurable simulation platform for recommender systems." arXiv preprint arXiv:1909.04847 (2019).

- [Koren et al., 2009] Yehuda Koren, Robert Bell, and Chris Volinsky. "Matrix factorization techniques for recommender systems." Computer 42, no. 8 (2009): 30-37.

- [Wang et al., 2017] Huazheng Wang, Qingyun Wu, and Hongning Wang. "Factorization bandits for interactive recommendation." In Proceedings of the AAAI Conference on Artificial Intelligence, vol. 31, no. 1. 2017.

---

### Decision · Program_Chairs · 2023-09-21

**Decision:**

Accept (poster)

**Comment:**

I am recommending an acceptance after a careful reading of the paper, reviews, and author rebuttal. In short, this is a well-written paper with sound results and a clear message – i.e., that refusing cookie consent makes it easier to infer demographics. My views on the submission broadly align with Reviewer [cPvy](https://openreview.net/forum?id=dFtpRphNb3&noteId=4ALdoUtjVm) who has recommended acceptance. I also share some (but not all) of the concerns raised by the other reviewers, and I would encourage the authors to address these issues in their revisions.

Looking forward, I would like to draw special attention to two particular concerns that multiple reviewers raised:

1. The fact that the paper supports their claim through a simple simulation model. In this case, the authors are correct that using a "real - model" would make it difficult to isolate effects. I would encourage the authors to explicitly discuss the inherent value of their approach in their revision as future readers will share it. I would also encourage the authors to include a discussion on their results to generalize to "real-world" recommender systems (which could include the experiments conducted during the rebuttal phase if need be).

2. The fact that the paper studies a problem but does not propose a concrete way to solve it. In this case, the authors are correct that it is worth drawing attention to problems that might not be easy to solve. This is a claim that does not merit special attention – though it does signal the need for a broader discussion on what can be done to resolve this issue.